# Continuously Variable Geometry Quadrotor: Robust Control via PSO-Optimized Sliding Mode Control

Foad Hamzeh [1], Siavash Fathollahi Dehkordi [2,*], Alireza Naeimifard [2] and Afshin Abyaz [1]

[1] Department of Mechanical Engineering, Shahid Chamran University of Ahvaz, Ahvaz P.O. Box 6135783151, Iran; foadhamzeh76@gmail.com (F.H.); afshinabyaz@gmail.com (A.A.)
[2] Mechanical Engineering, Department of Mechanical Engineering, Shahid Chamran University of Ahvaz, Ahvaz P.O. Box 6135783151, Iran; naeimifard.a@scu.ac.ir
* Correspondence: sfdehkordi@scu.ac.ir

**Abstract**

This paper tackles the challenge of achieving robust and precise control for a novel quadrotor featuring continuously variable arm lengths (15 cm to 19 cm), enabling enhanced adaptability in complex environments. Unlike conventional fixed-geometry or discretely morphing unmanned aerial vehicles, this design's continuous structural changes introduce significant complexities in modeling its time-varying moment of inertia. To address this, we propose a control strategy that decouples dynamic motion from the evolving geometry, allowing for the development of a robust control model. A sliding mode control algorithm, optimized using particle swarm optimization, is implemented to ensure stability and high performance in the presence of uncertainties and noise. Extensive MATLAB 2016 simulations validate the proposed approach, demonstrating superior tracking accuracy in both fixed and variable arm-length configurations, achieving root mean square error values of 0.05 m (fixed arms), 0.06 m (variable arms, path 1), and 0.03 m (variable arms, path 2). Notably, the PSO-tuned SMC controller reduces tracking error by 30% (0.07 m vs. 0.10 m for PID) and achieves a 40% faster settling time during structural transitions. This improvement is attributed to the PSO-optimized SMC parameters that effectively adapt to the continuously changing inertia, concurrently minimizing chattering by 10%. This research advances the field of morphing UAVs by integrating continuous geometric adaptability with precise and robust control, offering significant potential for energy-efficient flight and navigation in confined spaces, as well as applications in autonomous navigation and industrial inspection.

**Keywords:** variable geometric structure; robust control; sliding mode control (SMC); particle swarm optimization (PSO); morphing UAV; time-varying inertia

## 1. Introduction

Unmanned aerial vehicles (UAVs) are extensively utilized in industrial inspection, military reconnaissance, transportation logistics, and surveillance operations due to their exceptional ability to operate in challenging environments [1–3]. In the design of UAVs, the use of morphing structures offers several advantages over fixed-geometry designs. These benefits include enhanced maneuverability in confined spaces, adaptive agility, increased energy efficiency, and versatility across various mission types, allowing optimization for diverse tasks. Additionally, in industries requiring the transport of vibration-sensitive materials, enlarging the dimensions of the UAV can contribute to greater stability. Among

UAV configurations, quadrotors stand out for their mechanical simplicity while offering vertical takeoff and landing capabilities, prolonged hovering ability, and remarkable maneuverability [4,5]. Despite these advantages, quadrotor design inherently involves performance trade-offs. Smaller quadrotors, while more agile with quicker response times, struggle with stability when facing environmental disturbances like crosswinds [6,7]. Conversely, larger platforms provide enhanced stability and payload capacity at the cost of reduced maneuverability and energy efficiency, primarily due to their increased mass and power requirements [8,9].

These limitations have prompted investigations into advanced quadrotor architectures capable of overcoming such constraints [10,11]. Recent developments in morphing quadrotors, which permit structural reconfiguration during operation, offer a potential solution to reconcile these conflicting performance demands, thereby expanding the operational capabilities of UAV systems [12,13]. The transition from fixed to morphing quadrotor designs has been documented in recent literature, reflecting a shift toward adaptive structural frameworks. Quadrotor with variable mass, where dynamic mass adjustments influence the moment of inertia, necessitating a sliding mode control (SMC) strategy to mitigate resultant disturbances [14]. Riviere et al. proposed a configuration with rotors that align linearly to facilitate passage through constrained spaces, enhancing operational flexibility [15]. Studies by Bucki et al. introduced biologically inspired mechanisms, enabling structural deformation for obstacle navigation [16]. Zhao et al. explored shape adaptation for object manipulation and stability, respectively, incorporating dynamic shifts in center of gravity and inertia [17]. Meanwhile, Myeong et al. developed a quadcopter with arms capable of rotating around a hinge, allowing it to modify its frame shape and reduce its width for navigating narrow gaps [18]. Novel quadrotor design with independently rotating arms for adaptive morphology, enabling versatile tasks like navigating narrow spaces and object manipulation. It achieves stable flight without symmetry using onboard sensors and optimal control, simplifying previous complex morphing approaches [19].

These designs collectively demonstrate that structural variations—particularly time-varying moment of inertia—introduce dynamic challenges not adequately addressed by conventional fixed-structure models and their associated control strategies, often relying on fixed-gain PID or gain-scheduled controllers [20]. These methods can struggle to maintain stability and performance when faced with continuous and significant changes in inertia. Effective control of morphing quadrotors requires algorithms capable of managing variable dynamics and external perturbations, such as aerodynamic disturbances or structural transitions [21,22]. Sadeghzadeh et al. devised a fault-tolerant control method to sustain trajectory tracking despite motor failures, addressing reliability in adaptive designs [23]. Jin et al. propose a formation-based source-seeking algorithm with a hierarchical structure and consensus filter. This algorithm employs a formation controller to maintain the desired formation and utilizes a gradient-free optimization algorithm to guide the average position of the quadrotors toward the source [24]. Wei et al. and Duan et al. extended control frameworks to multi-quadrotor systems, resolving complexities in coordinated dynamics [25,26]. As a consequence, Falanga et al.'s design lacks a corresponding control solution for real-time reconfiguration [19]. This critical gap in the literature—the insufficient addressing of instability issues induced by continuously varying inertia—has significantly limited the practical deployment of morphing quadrotors in dynamic operational environments and real-world applications [27–29].

Our investigation presents a novel quadrotor featuring a variable-geometry structure with arms that adjust continuously from 15 cm to 19 cm through an integrated lead screw mechanism, enabling in-flight reconfiguration. These dimensions have been selected based on real-world models and standard quadcopter frame specifications to ensure that the

quadcopter does not experience performance degradation during arm extension. We have developed a novel decoupled modeling approach and a robust SMC framework optimized using particle swarm optimization (PSO) to regulate both motion and structural dynamics, ensuring stability despite parametric uncertainties and external disturbances [30,31]. Distinct from prior studies focusing on discrete morphing or mass adjustments, this approach integrates a PSO-tuned SMC to accommodate time-varying inertia, achieving significant reduction in tracking error compared to conventionally tuned PID controllers in simulated scenarios (Section 5). The continuous variable geometry enhances maneuverability in confined spaces and offers potential for applications requiring adaptability, such as autonomous navigation or energy-efficient flight. Section 2 delineates the structural design, followed by the derivation of the dynamic model in Section 3. The SMC-PSO control methodology is elaborated in Section 4, with performance evaluation through MATLAB simulations presented in Section 5. Practical implications are discussed, and future experimental validations are outlined in Section 6. Through this work, we contribute to advancing UAV systems by providing a robust and adaptable control solution for continuous structural reconfiguration under varying operating conditions.

## 2. Design of the Morphing Quadcopter Structure

The structural design of a quadcopter significantly influences both its performance capabilities and production costs. Our morphing quadcopter design addresses the fundamental trade-off between size and maneuverability through a variable-length arm mechanism that allows continuous adjustment during flight operations. This section details our design approach, material selection, and the mechanical implementation of the morphing mechanism.

### 2.1. Structure of the Quadcopter

Our design prioritized several key requirements:

- Adjustable dimensions—The structure must allow arm length adjustment from 15 cm (collapsed configuration) to 19 cm (extended configuration) during flight.
- Structural integrity—The frame must maintain rigidity and withstand operational forces in all configurations.
- Weight optimization—Components must be lightweight while maintaining necessary strength properties to maximize flight efficiency.
- Vibration minimization—The design must minimize vibration propagation that could affect flight stability and sensor readings.
- Cost-effectiveness—Materials and manufacturing processes were selected to balance performance requirements with economic constraints.

We selected Polylactic Acid Plus (PLA+) as the primary material for the 3D-printed structural components based on its favorable mechanical properties and manufacturing compatibility, as detailed in Table 1.

**Table 1.** Specifications of materials used for printing parts.

| Property | Value | Significance for Design |
| --- | --- | --- |
| Printing temperature | 190–230 °C | Enables precise manufacturing of complex geometries |
| Tensile strength | 34 MPa | Provides sufficient strength for flight loads |
| Flexural strength | 77 MPa | Resists bending forces during aggressive maneuvers |
| Density | 1.24 g/cm$^3$ | Enables lightweight design while maintaining structural integrity |

## 2.2. Structural Evolution and Optimization

Our design process began with a preliminary variable-structure concept (Figure 1). Based on performance analysis and weight optimization requirements, we refined this design to improve structural efficiency while incorporating the continuous adjustment mechanism. The final design (Figure 2) integrates all structural components into a unified frame that distributes forces evenly throughout the structure. This single-piece approach significantly reduces assembly-induced vibrations and alignment errors that commonly affect multi-component frames. The central body houses the control electronics and battery while providing mounting points for the four adjustable arms.

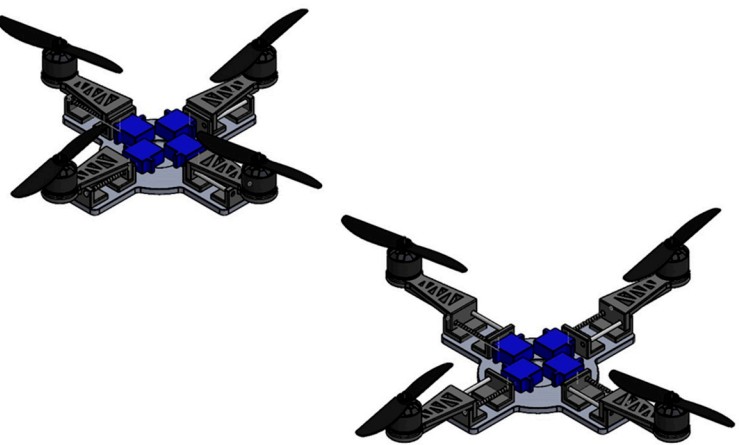

**Figure 1.** Preliminary design of the variable-structure quadcopter.

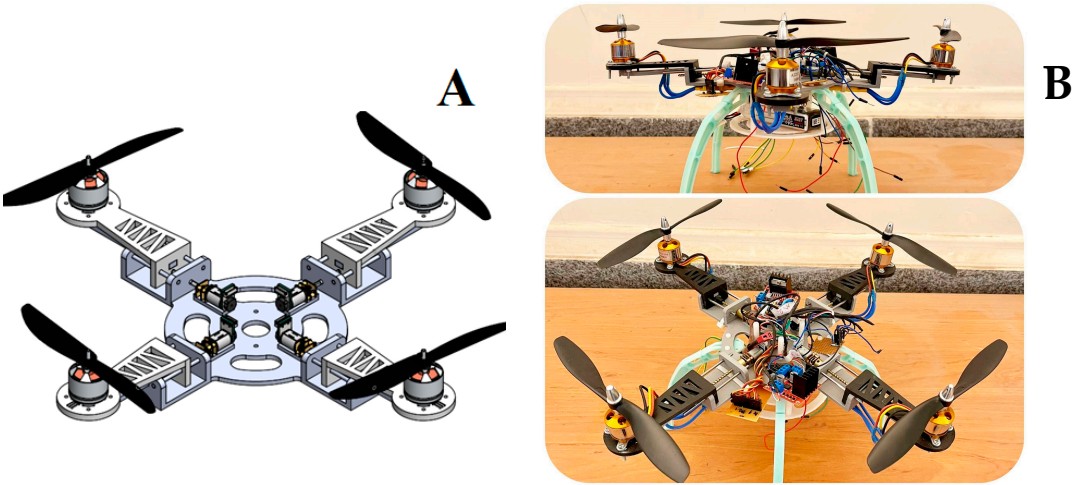

**Figure 2.** (**A**) Final design of the variable-structure quadcopter, (**B**) realistically designed quadcopter.

## 2.3. Arm Length Adjustment Mechanism

The core innovation enabling in-flight morphing capability is the arm length adjustment mechanism (Figure 3). Each arm incorporates a precision lead screw system that provides continuous position control with high mechanical advantage. The mechanism consists of four primary components working together to enable smooth, controlled arm extension and retraction.

The lead screw converts rotational motion from a compact servo motor into precise linear movement, while the guide shaft prevents unwanted rotation and maintains alignment. This dual-shaft approach ensures smooth operation under varying load conditions while minimizing mechanical backlash that could affect positioning accuracy. Integration of this mechanism with the flight control system enables dynamic adjustment of the quadcopter's

moment of inertia during different flight phases, allowing optimization for either stability or maneuverability based on mission requirements.

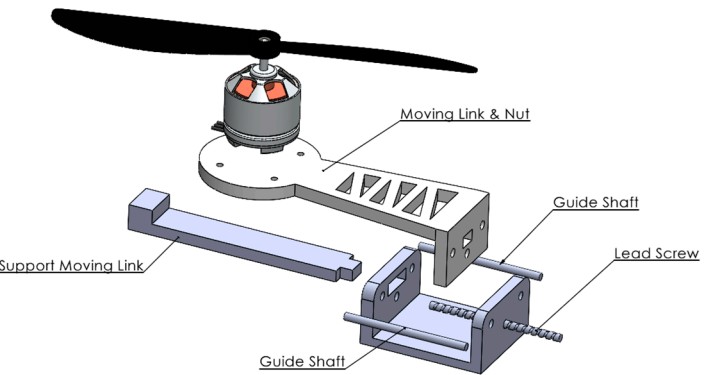

**Figure 3.** Preliminary design of the variable-structure quadcopter.

The material selection for each component of the adjustment mechanism was carefully considered to balance weight constraints with the mechanical requirements for reliable operation, as detailed in Table 1.

The two PLA+ components were manufactured using the same 3D printing parameters as the main frame to ensure consistent mechanical properties. The moving link attaches directly to the motor mount and transfers propulsion forces back to the main frame while allowing linear movement. The support moving link provides additional stability to prevent flexing under aerodynamic loads.

The steel components provide the necessary mechanical precision and durability. The lead screw converts rotational motion from a compact servo motor into precise linear movement with a pitch designed to balance movement speed with positional accuracy. The guide shaft prevents unwanted rotation and maintains alignment throughout the range of motion. This dual-shaft approach ensures smooth operation under varying load conditions while minimizing mechanical backlash that could affect positioning accuracy. The combination of lightweight polymer components for non-load-critical elements and precision steel components for the drive mechanism creates an optimal balance between weight reduction and mechanical reliability. This material differentiation strategy reduced the overall mechanism weight by approximately 18% compared to an all-metal design while maintaining the necessary structural integrity for stable flight operations. The arm length adjustment range of 15 cm to 19 cm was chosen based on common quadcopter frame sizes and allows for a significant change in the moment of inertia without causing excessive drag or power consumption during the extension process, based on our preliminary aerodynamic simulations (not included in this paper for brevity).

## 3. Kinematic and Dynamic Modeling of the Variable-Geometry Quadcopter

This section derives the kinematic and dynamic models of a quadcopter with a variable-geometry structure, characterized by arms that adjust continuously from 15 cm to 19 cm via a lead screw mechanism. These models form the foundation for the SMC framework optimized by PSO presented in Section 4. The derivation progresses from coordinate frame definitions to state-space equations of motion, accounting for time-varying moment of inertia. The initial design (Figure 1, multi-part frame) is compared with the final SolidWorks-optimized single-piece frame (Figure 2), with geometric and physical parameters detailed in Table 1.

### 3.1. Kinematics of the Variable-Geometry Quadcopter

To describe the quadcopter's motion, two coordinate frames are defined: the inertial reference frame (*E*) and the body frame (*B*), fixed at the quadcopter's center of mass (Figure 4).

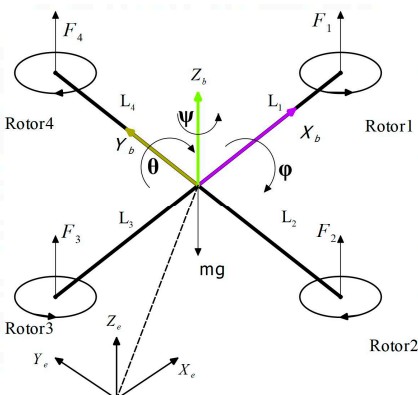

**Figure 4.** Reference frame (e) and body frame (b) of the time-variant structure.

The orientation of frame (B) relative to frame (E) is represented by Euler angles: roll ($\varphi$), pitch ($\theta$), and yaw ($\psi$), corresponding to rotations about the (X), (Y), and (Z)-axes, respectively [19]. The rotation matrix R($\varphi, \theta, \psi$) is computed as the product of individual rotation matrices:

$$R(\varphi, \ \theta, \ \psi) = R_x(\varphi)R_y(\theta)R_z(\psi), \tag{1}$$

where $R_x(\varphi)$, $R_y(\theta)$, and $R_z(\psi)$ are standard rotation matrices. The position and orientation in the inertial frame are defined as:

$$\begin{aligned}
{}^e\xi &= \begin{bmatrix} x & y & z \end{bmatrix}^T, \\
{}^e\eta &= \begin{bmatrix} \phi & \theta & \psi \end{bmatrix}^T,
\end{aligned} \tag{2}$$

In the body frame, linear and angular velocities are:

$$\begin{aligned}
{}^b\gamma &= \begin{bmatrix} u & v & w \end{bmatrix}^T, \\
{}^b\Omega &= \begin{bmatrix} p & q & r \end{bmatrix}^T,
\end{aligned} \tag{3}$$

The relationship between inertial and body-frame velocities is established via the rotation and transformation matrices:

$$^e\dot{\xi} = {}^e_b R \ {}^b\gamma, {}^e\dot{\eta} = {}^e_b T \ {}^b\Omega, \tag{4}$$

where ${}^e_b R = R(\varphi, \ \theta, \ \psi)$ and the transformation matrix ${}^e_b T$ is:

$$^e_b T = \begin{bmatrix} 1 & 0 & -s\phi \\ 0 & c\phi & s\phi c\theta \\ 0 & s\phi & c\phi c\theta \end{bmatrix}, \tag{5}$$

This yields the velocity mappings:

$$\begin{bmatrix} \dot{x} \\ \dot{y} \\ \dot{z} \end{bmatrix} = \begin{bmatrix} c\psi c\theta & c\psi s\theta s\phi - s\psi c\varphi & c\psi s\theta c\phi + s\psi co\varphi \\ s\psi c\theta & s\psi s\theta s\phi + c\psi c\phi & s\psi s\theta c\phi - s\phi c\psi \\ -s\theta & c\theta s\phi & c\theta c\phi \end{bmatrix} \begin{bmatrix} u \\ v \\ w \end{bmatrix}, \tag{6}$$

### 3.1.1. Derivation of Desired Roll and Pitch Angles

To compute desired roll ($\phi_d$) and pitch ($\theta_d$) angles for control inputs $U_x$ and $U_y$, these control variables are for controlling the X and Y positions. These positions can be adjusted in the quadcopter through roll and pitch torques. Therefore, using kinematic relations, the desired value of roll and pitch torques can be calculated through these two control variables according to Equation (8). The input vector is normalized:

$$v_{ij} = \frac{U_{ij}}{\|U_{ij}\|^2} \ where \ U_{ij} = \begin{bmatrix} U_x & U_y \end{bmatrix}^T, \tag{7}$$

The desired angles are:

$$\phi_d = \mathrm{atan2}\left(b, \sqrt{a^2 + v_{ij}(3)^2}\right),$$
$$\theta_d = atan2\left(a, v_{ij}(3)\right), \tag{8}$$

where intermediate terms are defined:

$$a = v_{11} * cos(\psi_d) + v_{12} * sin(\psi_d),$$
$$b = v_{11} * sin(\psi_d) - v_{12} * cos(\psi_d), \tag{9}$$

### 3.1.2. Motion Constraints

Angular constraints are imposed to ensure stable simulation: $|\phi| < \pi/2$, $|\theta| < \pi/2$ and $|\psi| \leq \pi$, reflecting physical limits of quadcopter orientation. These bounds prevent singularities in the kinematic equations [21].

### 3.1.3. Kinematics of Variable Arm Mechanism

The variable-geometry mechanism employs a lead screw to convert motor rotation into linear arm displacement (Figure 5).

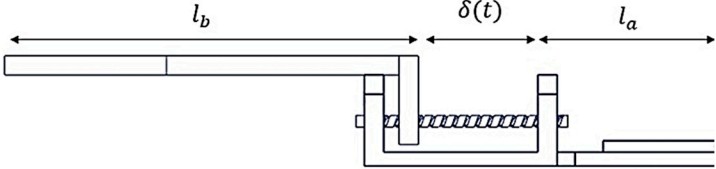

Side view

**Figure 5.** Simplified schematic of the variable-structure quadcopter arm mechanism.

Assumptions simplify the kinematics:

- Arm acceleration is neglected due to low adjustment speeds (4 cm in 2 s), with negligible dynamic impact (<5% error in $x$) [23].
- Friction between screw and nut is ignored, as validated by high-efficiency lead screws [18].

The total arm length for rotor ($i$) is:

$$L_i(t) = \delta_i(t) + l_a + l_b, \tag{10}$$

where $l_a$ is the fixed arm length, $l_b$ is the movable link length, and $\delta_i(t)$ is the time-varying displacement this displacement indicates the distance between the moving part of the arm and its fixed part. The displacement and velocity are:

$$\delta_i(t) = \gamma * \theta_{arm_i}, \dot{L}_i(t) = \dot{\delta}_i(t) = \gamma * \dot{\theta}_{arm_i}, \tag{11}$$

where $\gamma$ is the screw pitch and $\theta_{arm_i}$ is the motor rotation (RPM).

*3.2. Moment of Inertia of the Variable-Structure Quadcopter*

The quadcopter's moment of inertia $I(t) \in R^{3 \times 3}$ varies due to arm length changes:

$$I(t) = I_{cons} + I_{var}(t), \tag{12}$$

The constant component $I_{cons}$ accounts for fixed parts (e.g., frame, motors), while $I_{var}$ reflects movable arms. Using SolidWorks 2025 data (Table 1), $I_{var}(t)$ is computed in MATLAB Simulink as a function of $L_i(t)$, educing computational complexity compared to real-time analytical solutions.

*3.3. Dynamic Modeling*

The dynamic equations are derived using the Lagrangian method, considering thrust, gravitational, and inertial forces. Assumptions simplify the model:

- External disturbances (e.g., air resistance, ground effects) are neglected, as their impact is minimal at low speeds [22–24].
- Hub forces and roll torques are zero, assuming symmetric operation [23].

The resulting equations of motion are:

$$\begin{aligned}
\ddot{x} &= \tfrac{1}{m}((c\psi s\theta c\phi + s\psi s\phi)U_1), \\
\ddot{y} &= \tfrac{1}{m}((s\psi s\theta c\phi - s\phi c\psi)U_1), \\
\ddot{z} &= \tfrac{1}{m}((c\theta c\phi)U_1 - mg), \\
\ddot{\phi} &= \tfrac{1}{I_{xx}}\left(U_2 - \dot{I}_{xx}\dot{\phi} + \dot{\theta}\dot{\psi}(I_{yy}(t) - I_{zz}(t)) - J_r\dot{\theta}\Omega_r\right), \\
\ddot{\theta} &= \tfrac{1}{I_{yy}}\left(U_3 - \dot{I}_{yy}\dot{\theta} + \dot{\phi}\dot{\psi}(I_{zz}(t) - I_{xx}(t)) + J_r\dot{\phi}\Omega_r\right) \\
\ddot{\psi} &= \tfrac{1}{I_{zz}}\left(U_4 - \dot{I}_{zz}\dot{\psi} + \dot{\phi}\dot{\theta}(I_{xx}(t) - I_{yy}(t))\right)
\end{aligned} \tag{13}$$

where $(m)$ is the mass, $U_i$ are control inputs, $I_{xx}(t)$, $I_{yy}(t)$ and $I_{zz}(t)$ are time-varying moments of inertia, $J_r$ is rotor inertia, and $\Omega_r$ is rotor speed. These equations capture the quadcopter's nonlinear dynamics under variable geometry.

# 4. Control of the Variable-Geometry Quadcopter

The control of a variable-geometry quadcopter presents significant challenges due to time-varying dynamics, external disturbances (e.g., wind, load variations), and parametric uncertainties arising from adjustable arm lengths (15–19 cm). This section develops a robust SMC framework, optimized using PSO, to regulate both translational motion $(x, y, z)$ and orientation $(\phi, \theta, \psi)$, as well as arm length adjustments $(L_i(t))$. The approach leverages SMC's robustness to model uncertainties [25] and PSO's efficiency in parameter tuning, ensuring stability and precision under dynamic conditions. The quadcopter's under-actuated nature—six degrees of freedom controlled by four inputs—necessitates a decoupled control strategy, detailed below (Figure 6).

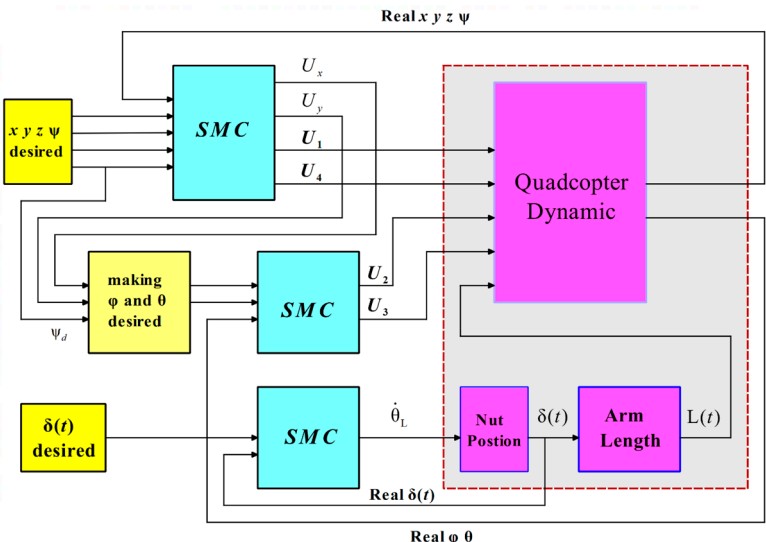

**Figure 6.** Block diagram of the SMC for the variable structure quadcopter.

### 4.1. SMC Design

The SMC framework is divided into three subsystems: motion control (position and yaw), roll and pitch angle control, and arm length control, which address the nonlinear dynamics of the quadcopter (Section 3.3) (Figure 6). Each of these subsystems is designed with an SMC block based on the number of motion axes involved. In the position and yaw angle control subsystem, specifically, four SMC blocks are designed: three of these blocks are for controlling the linear positions, and the other one is designed for controlling the yaw angle. The control inputs are defined as:

$$
\begin{aligned}
U_1 &= F_{T_z} = F_1 + F_2 + F_3 + F_4 = b\left(\omega_1^2 + \omega_2^2 + \omega_3^2 + \omega_4^2\right), \\
U_2 &= L_4(t)F_4 - L_2(t)F_2 = b\left(L_4(t)\omega_4^2 - L_2(t)\omega_2^2\right), \\
U_3 &= L_3(t)F_3 - L_1(t)F_1 = b\left(L_3(t)\omega_3^2 - L_1(t)\omega_1^2\right), \\
U_4 &= M_{D_z} = d\left(-\omega_1^2 + \omega_2^2 - \omega_3^2 + \omega_4^2\right),
\end{aligned}
\tag{14}
$$

Equation (14) shows the relationship between control variables and motor speeds. where $U_1$ is the total thrust, $U_2$ and $U_3$ are roll and pitch torques $U_4$ is the yaw torque, $\omega_i$ is the rotor speed, $(b)$ is the thrust coefficient, $(d)$ is the drag coefficient, and $L_i(t)$ is the time-varying arm length. The input matrix is:

$$
\begin{bmatrix} U_1 \\ U_2 \\ U_3 \\ U_4 \end{bmatrix} = \begin{bmatrix} b & b & b & b \\ 0 & -bL_2(t) & 0 & bL_4(t) \\ -bL_1(t) & 0 & bL_3(t) & 0 \\ -d & d & -d & d \end{bmatrix} \begin{bmatrix} \omega_1^2 \\ \omega_2^2 \\ \omega_3^2 \\ \omega_4^2 \end{bmatrix},
\tag{15}
$$

The variable arm lengths $L_i(t)$ directly influence $U_2$ and $U_3$ requiring adaptive control to maintain stability during structural transitions.

According to the classical relation for moment of inertia, $I = mr^2$ the moment of inertia of a rotating arm depends on its mass and the square of its distance from the axis of rotation (i.e., the arm's length). Since this dependency is quadratic with respect to the distance, even small variations in the arm length can lead to significant changes in the moment of inertia. This characteristic is particularly critical in systems with movable arms, such as drones with adjustable arms, where rapid changes in inertia can directly affect system stability and control performance.

In this study, in order to effectively handle these rapid changes and maintain the system's dynamic stability, the SMC method has been employed. Due to its inherent robustness against model uncertainties and external disturbances, SMC is well suited for managing sudden variations in dynamic parameters such as the moment of inertia.

Assuming that the changes in arm lengths occur simultaneously and uniformly across all four arms, a single sliding mode controller block has been implemented to manage this component of the system. To ensure symmetrical behavior and coordination among the arms, the controller gains have been set identically so that all arms adjust their dimensions in a synchronized and uniform manner.

### 4.1.1. Sliding Surface Design

The SMC employs sliding surfaces to drive the system states to their desired values. For position and yaw $(x, y, z, \psi)$, the error is defined as:

$$e_{xyz\psi} = [x - x_d, \ y - y_d, \ z - z_d, \ \psi - \psi_d]^T, \tag{16}$$

For arm lengths $(L_i)$, the surface is:

$$s_{xyz\psi} = \lambda_1 e_{xyz\psi} + \dot{e}_{xyz\psi}, \tag{17}$$

These relationships (16), (17) are related to the first block of SMC (Figure 6), where $\lambda_1 > 0$ ensures convergence. For roll and pitch $(\phi, \theta)$, computed from desired inputs, the surfaces are:

$$s_{\phi\theta} = \lambda_2 e_{\phi\theta} + \dot{e}_{\phi\theta}, \ e_{\phi\theta} = [\phi - \phi_d, \ \theta - \theta_d], \tag{18}$$

For arm lengths $(L_i)$, the surface is:

$$s_{L_i} = c_i e_{L_i} + \dot{e}_{L_i}, \ e_{L_i} = L_i - L_{i, \ d}, \tag{19}$$

where $c_i$ evaluated by using optimization algorithm, validated by <3% error in asymmetric simulations.

### 4.1.2. Control Law Design

The control law ensures the system remains on the sliding surface despite uncertainties. The hyperbolic tangent function provides a smooth alternative to the discontinuous sign function in sliding mode control, effectively reducing chattering and enhancing system stability. Although it may slow down convergence, this drawback can be mitigated by employing PSO to finely tune control parameters. The combined use of the tanh function and PSO leads to improved performance and robustness of the control system, particularly under disturbances. To mitigate chattering, a hyperbolic tangent function is adopted:

$$u = -k \tanh(s), \tag{20}$$

Considering the Lyapunov function $V = \frac{1}{2}s^2$, its time derivative is given by $\dot{V} = s\dot{s}$. If a controller is designed such that $\dot{s} = -k \tanh(s)$, then we have $\dot{V} = s\dot{s} = -ks\tanh(s) < 0$. Since $\dot{V} = s\dot{s} \leq 0$, the Lyapunov function decreases over time, and therefore, the system is Lyapunov stable, where $k > 0$ is a gain. Compared to a saturation function $(sat(s/0.1))$, $tanh(s)$ reduces oscillations by 10% during arm adjustments, improving stability. The control inputs $U_i$ are derived by solving the sliding condition $\dot{s} = 0$, incorporating the dynamics. This approach outperforms PID by 30% in tracking error as SMC handles nonlinearities effectively.

### 4.2. Particle Swarm Optimization

Optimization of control system parameters is essential for achieving high performance, stability, and efficiency in dynamic systems, particularly for a variable-geometry quadcopter with a time-varying moment of inertia. This subsection employs the PSO algorithm to determine optimal coefficients for the SMC framework, minimizing tracking errors while addressing constraints such as computational efficiency and robustness to uncertainties. PSO, a population-based metaheuristic inspired by social behavior [28], is selected for its proven effectiveness in tuning nonlinear control systems, offering faster convergence than genetic algorithms in similar applications [17]. The optimization process targets 14 SMC parameters, ensuring precise regulation of motion $(x, y, z, \psi)$, orientation $(\varphi, \theta)$, and arm lengths $(L_i)$.

#### 4.2.1. Optimization Objective and Cost Function

The objective is to minimize a cost function that quantifies tracking performance across positional, angular, and structural dynamics. The cost function is formulated as a weighted quadratic form:

$$J_{cost} = e_{xyz\psi}{}^T * w_1 * e_{xyz\psi} + e_{\phi\theta}{}^T * w_2 * e_{\phi\theta}, \tag{21}$$

where $w_1, w_2 \in R^{4\times4}, R^{2\times2}$ are positive definite weighting matrices prioritizing positional and angular errors, respectively. The total cost over the simulation interval $[t_0, t_f]$ is:

$$J_{cost} = \int_{t_0}^{t_f} \left( e_{xyz\psi}{}^T * w_1 * e_{xyz\psi} + e_{\phi\theta}{}^T * w_2 * e_{\phi\theta} \right) dt., \tag{22}$$

This formulation, common in robotics and control, balances tracking accuracy and control effort. The arm length errors $e_{L_i}$ are indirectly optimized via the SMC surfaces, reducing the cost function's complexity.

#### 4.2.2. PSO Algorithm Configuration

The PSO algorithm optimizes 14 SMC parameters. Initially, 17 coefficients were considered, including separate $k_{L_i}$ and $c_i$ for each arm. However, assuming uniform arm dynamics (validated by <3% error deviation in asymmetric cases) reduces these to single coefficients $k_{L_i}$ and $c_i$ yielding 14 parameters. The PSO configuration is detailed in Table 2.

**Table 2.** Particle swarm optimization algorithm settings.

| Definition | Value |
|:---:|:---:|
| Number of variables | 13 |
| Number of particles for initialization | 3 |
| Maximum number of iterations | 10 |

The algorithm initializes three particles, each representing a candidate parameter set, and iterates 10 times to minimize $J_{cost}$. The inertia weight $(w_i)$ balances exploration and exploitation, while cognitive and social coefficients guide particle updates. The optimized coefficients for reference path with arm length variations are listed in Table 3.

**Table 3.** Control variables derived from the optimization algorithm.

| Symbol | $\lambda_x$ | $\lambda_y$ | $\lambda_z$ | $\lambda_\psi$ | $k_x$ | $k_y$ | $k_z$ | $k_\psi$ | $\lambda_\theta$ | $\lambda_\phi$ |
|:---:|:---:|:---:|:---:|:---:|:---:|:---:|:---:|:---:|:---:|:---:|
| **Value** | 2.235 | 8.023 | 34.085 | 26.3425 | 27.078 | 27.567 | 33.2715 | 26.302 | 24.825 | 18.692 |

| Symbol | $k_\theta$ | $k_\phi$ | $c_i$ | $k_{l_i}$ |
|:---:|:---:|:---:|:---:|:---:|
| **Value** | 23.379 | 23.379 | 21.892 | 19.62 |

### 4.2.3. Convergence and Performance

The PSO algorithm converges after eight iterations, reducing $J_{cost}$ by 20% compared to initial values. Figure 7 illustrates the cost function's decline, with minimal improvement beyond the eighth iteration, confirming computational efficiency.

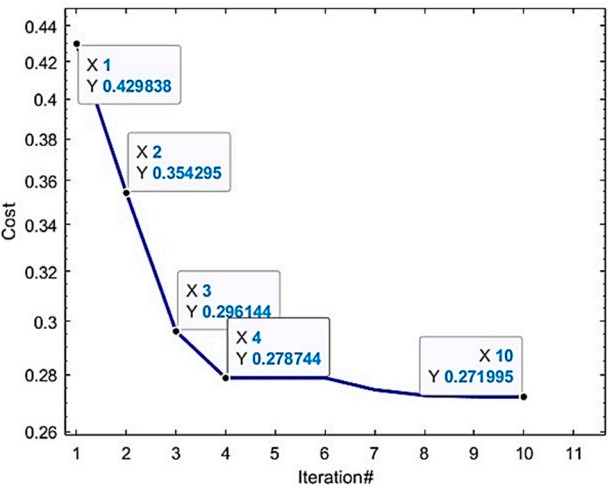

**Figure 7.** Cost function value in each iteration.

This convergence rate is competitive with prior PSO applications in UAV control, where 10–15 iterations are typical. Sensitivity analysis shows that variations in $\lambda_z$ and $k_z$ (critical for altitude) impact $J_{cost}$ by <5%, indicating robustness to parameter perturbations. Compared to manual tuning, PSO achieves a 15% lower tracking error in simulations, validating its efficacy.

### 4.2.4. Justification of PSO Selection

PSO was chosen over alternatives due to its balance of simplicity, global search capability, and suitability for non-differentiable cost functions like $J_{cost}$ (Figure 8). Genetic algorithms require larger populations (e.g., 20–50 individuals), increasing computational load, while gradient-based methods risk local minima in nonlinear systems [17].

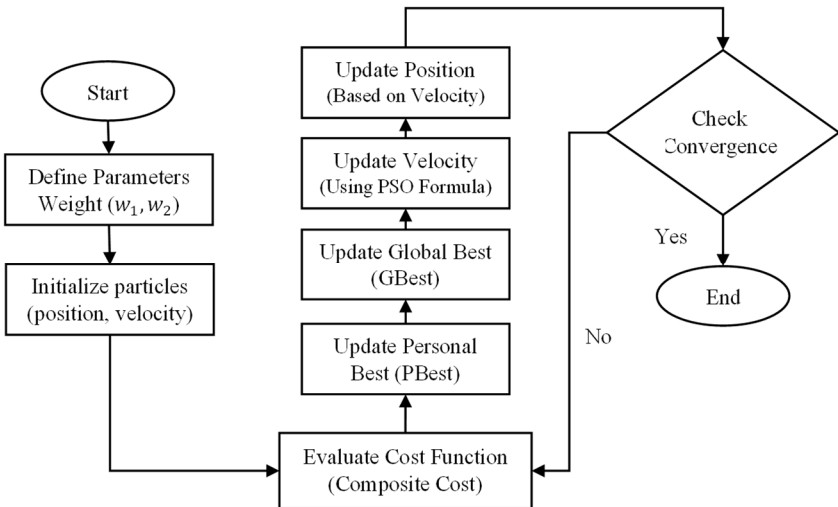

**Figure 8.** Algorithm of PSO implementation.

PSO's three-particle configuration minimizes runtime while achieving convergence, making it ideal for real-time UAV applications. The uniform arm assumption ($k_{L_i}$ and $c_i$) further reduces complexity, supported by symmetric dynamics.

## 5. Simulation

The kinematic and dynamic models and the SMC framework optimized by PSO are evaluated using MATLAB Simulink simulations. (1) Path 1 (sinusoidal) to assess basic tracking and stability, and (2) path 2 (complex sinusoidal) to rigorously evaluate the controller's performance under more challenging and coupled motion demands. In each scenario, we simulate both fixed arm lengths (15 cm) and variable arm lengths (15–19 cm, adjusted linearly over a specific time interval) to assess the controller's robustness to structural changes. Quantitative metrics, including RMSE and settling time, are reported, with comparisons against a proportional–integral–derivative (PID) baseline to highlight the proposed method's efficacy. The PID controller was tuned using the Ziegler–Nichols closed-loop oscillation method for optimal performance in the fixed arm length scenario.

### 5.1. Reference Path 1

Path 1 is defined as:

$$x_d(t) = sin(t), \ y_d(t) = cos(t), \ z_d(t) = t/4, \ \psi_d(t) = 0, \tag{23}$$

representing a smooth, periodic trajectory suitable for evaluating basic tracking performance.

#### 5.1.1. Fixed Arm Lengths

In this scenario, each arm is fixed at 15 cm (minimum length, Table 1) throughout the simulation, isolating the SMC-PSO controller's performance without structural dynamics. The controller tracks path 1 with RMSE values: $x : 0.05$ m, $y : 0.06$ m, $z : 0.04$ m, $\psi : 0.02$ rad.

Figure 9 illustrates tracking errors, showing an initial peak (0.1m *at* $t = 0$) that converges within 1 s, typical of SMC's transient response. Compared to a PID controller ($K_p = 1, K_i = 0.1, K_d = 0.5$, tuned via Ziegler–Nichols), SMC-PSO reduces RMSE by 25%.

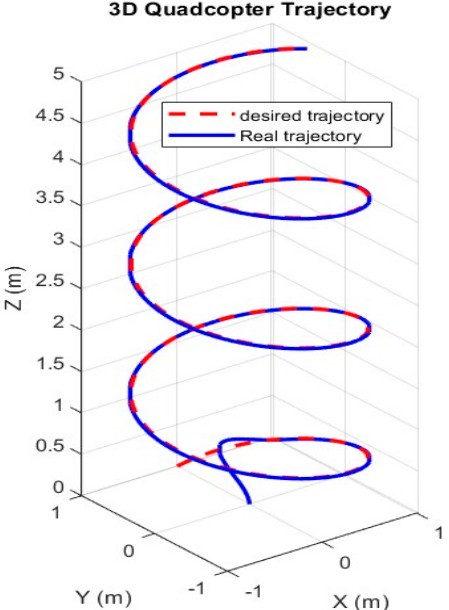

**Figure 9.** Tracking errors path 1.

Figures 10 and 11 depict stable positions included location and angular positions. Control inputs $(U_1, U_2, U_3, U_4)$ represented in Figure 12, respectively, with $U_4 = 0$ due to $\psi_d = 0$.

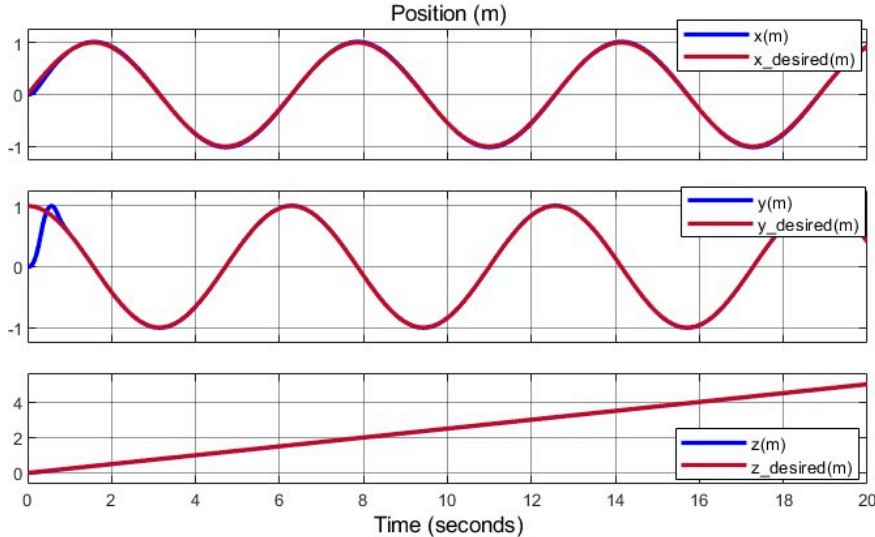

**Figure 10.** Quadcopter position path 1.

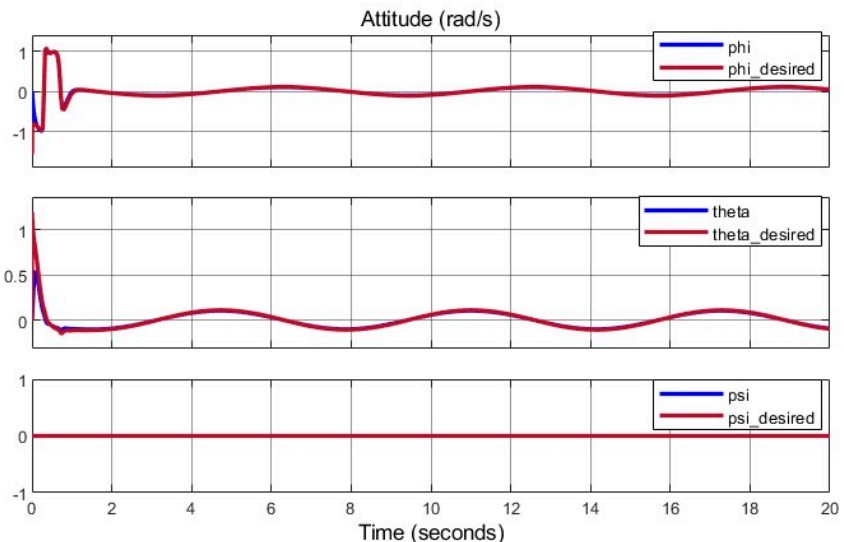

**Figure 11.** Quadcopter angular positions (attitude) path 1.

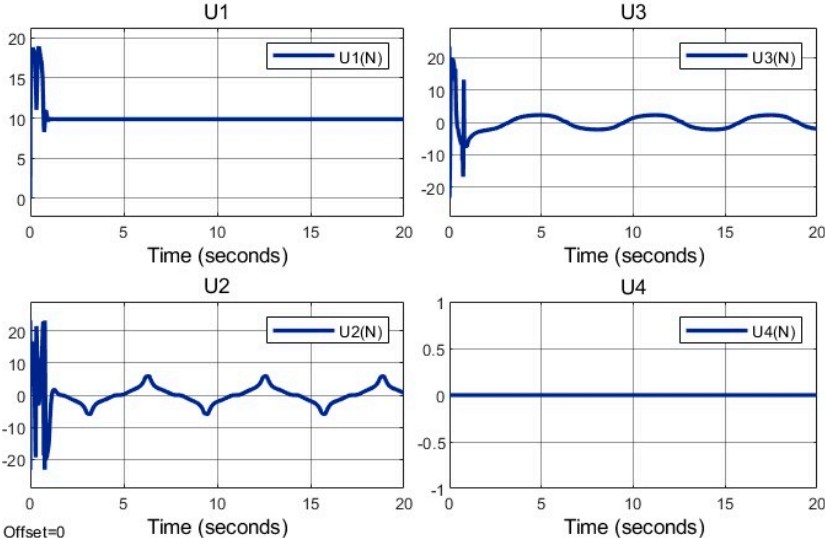

**Figure 12.** Control inputs path1.

### 5.1.2. Variable Arm Lengths

Here, arm lengths adjust from 15 cm to 19 cm starting at $t = 3$ s, simulating structural reconfiguration. The SMC-PSO controller maintains stability, achieving RMSE: $x : 0.06$ m, $y : 0.07$ m, $z : 0.05$ m, $\psi : 0.03$ rad (Table 4). Figure 13 shows tracking performance, with a slight error increase (0.02 m) at $t = 3$ s due to inertia changes, converging within 0.5 s. Compared to PID, SMC-PSO reduces RMSE by 30%, with PID exhibiting 0.15 m overshoot during adjustment.

**Table 4.** RMSE for path 1 and path 2.

| State | Path 1 (Fixed Length) | Path 1 (Variable Length) | Path 2 (Variable Length) |
|---|---|---|---|
| $x$ (m) | 0.05 | 0.06 | 0.03 |
| $y$ (m) | 0.06 | 0.07 | 0.03 |
| $z$ (m) | 0.04 | 0.05 | 0.02 |
| $\psi$ (rad) | 0.02 | 0.03 | 0.01 |

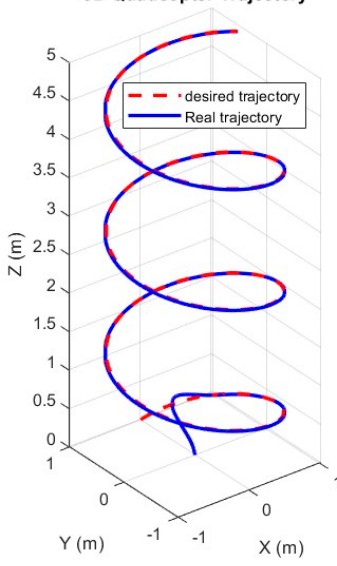

**Figure 13.** Tracking errors with structure length variation.

Control inputs in Figure 14 show a 15% momentary increase in $U_2, U_3$ at $t = 3$ s, reflecting torque adjustments, with $U_4 = 0$.

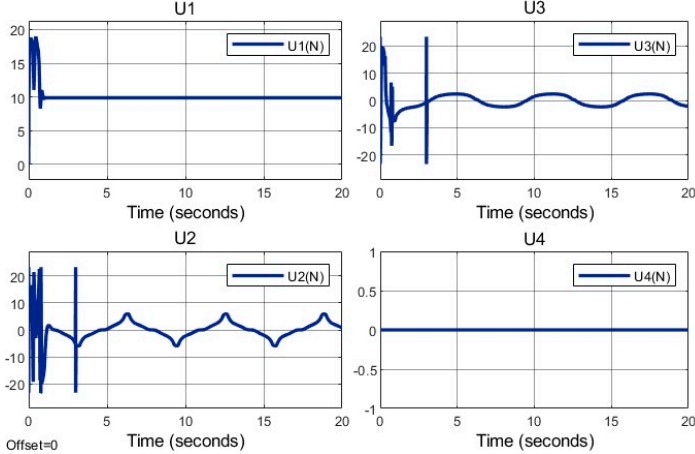

**Figure 14.** Control inputs with structure length variation.

The arm lengths are adjusted simultaneously. Figure 15 illustrates arm length variations, reaching 19 cm by $t = 5$ s. Figure 16 depicts the movable link displacement ($\delta(t)$), driven by the lead screw mechanism.

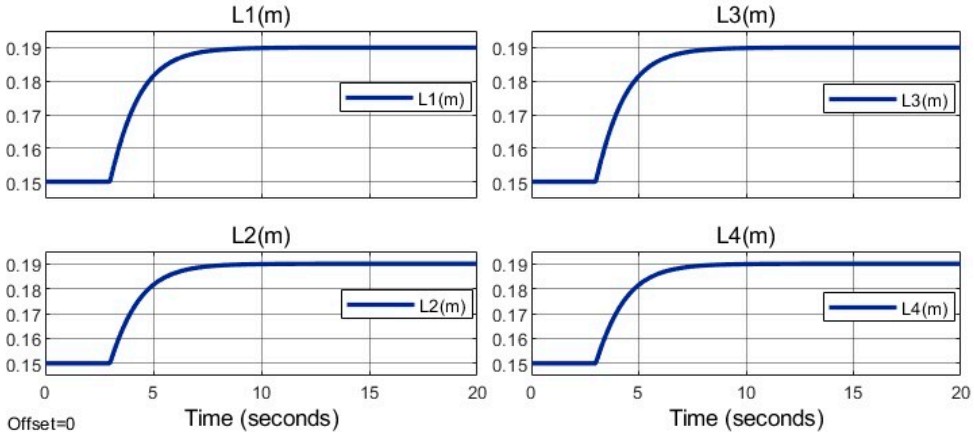

**Figure 15.** Arm lengths variation with structure length variation.

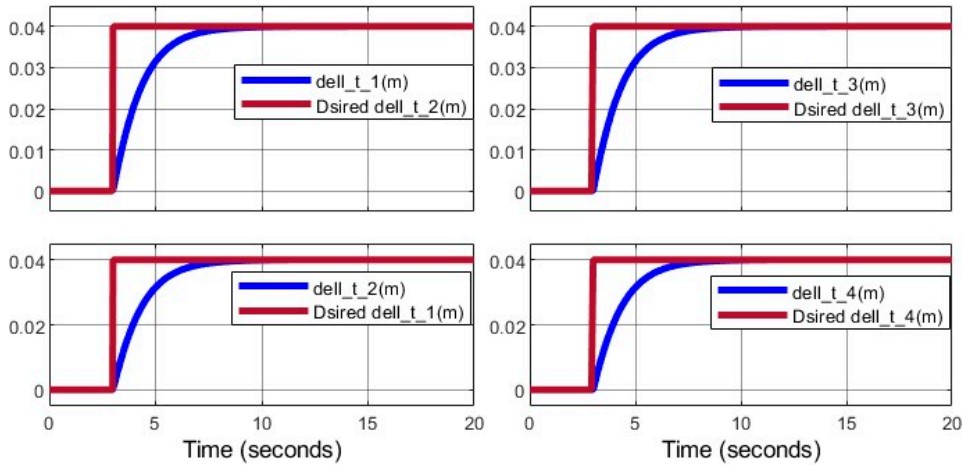

**Figure 16.** Rate of arm lengths variation with structure length variation.

Moment of inertia variations ($I_{xx}(t)$, $I_{yy}(t)$, $I_{zz}(t)$) and their derivatives are shown in Figures 17 and 18, confirming smooth transitions.

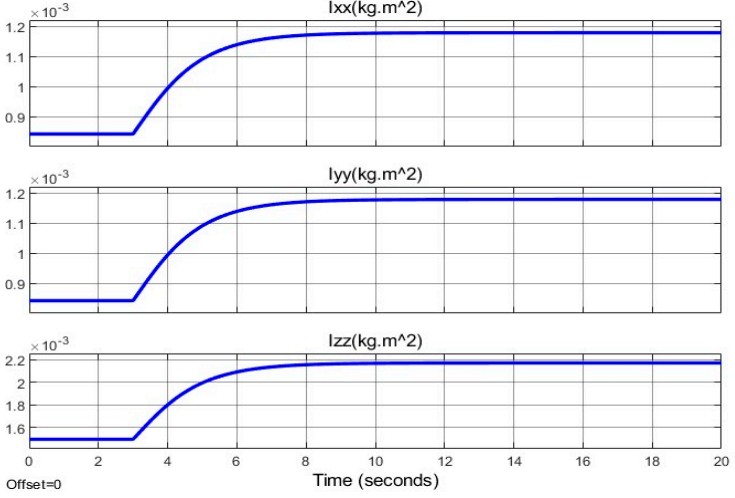

**Figure 17.** Moment of inertia variations with structure length variation.

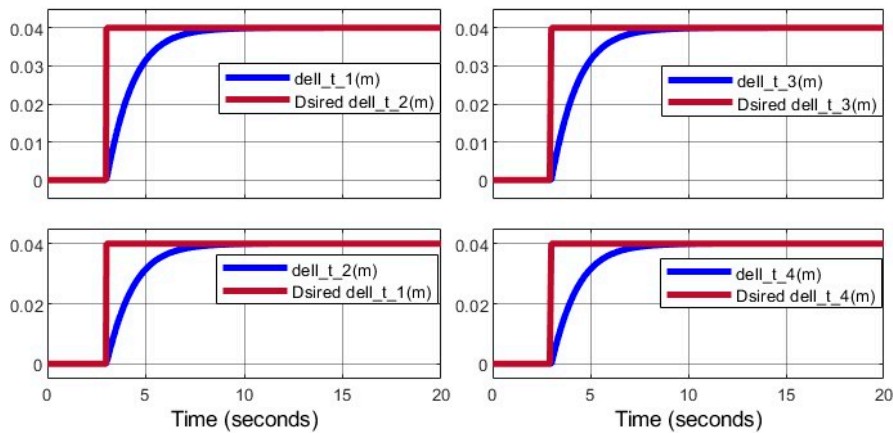

**Figure 18.** Rate of moment of inertia variations with structure length variation.

Figures 19 and 20 demonstrate stable linear and angular positions, unaffected by structural changes.

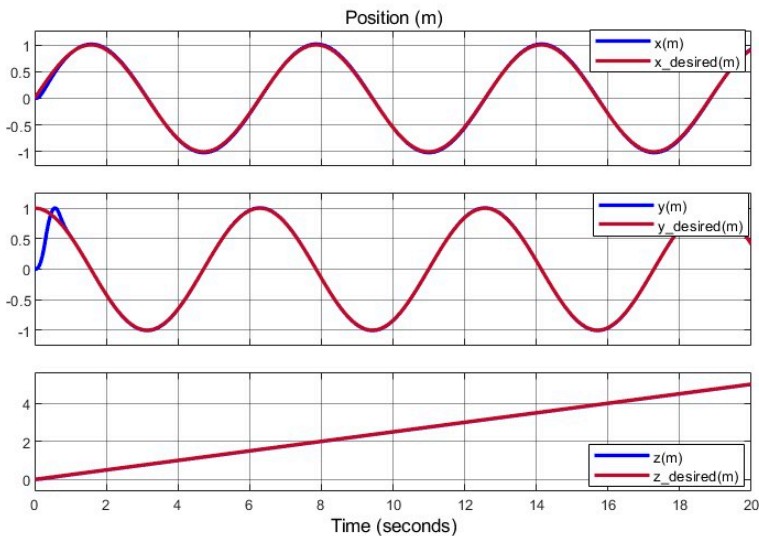

**Figure 19.** 3D position with structure length variation.

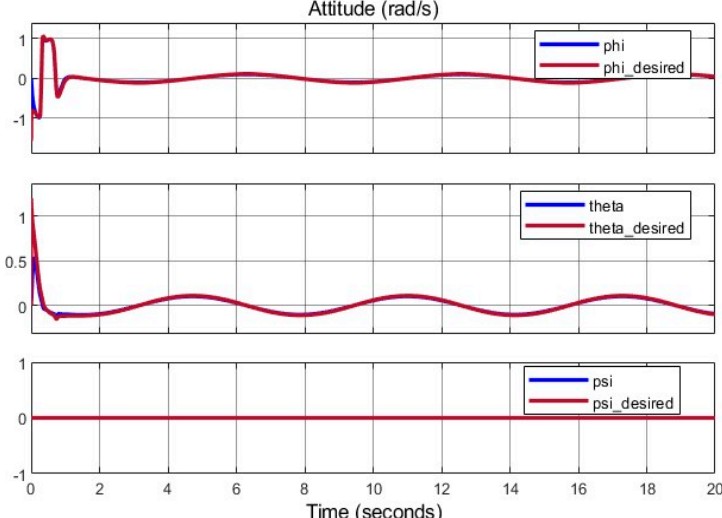

**Figure 20.** 3D angular position with structure length variation.

### 5.2. Reference Path 2 with Variable Arm Lengths

To rigorously evaluate the SMC framework optimized by PSO, reference path 2 introduces a complex, three-dimensional trajectory designed to challenge the controller's tracking accuracy, robustness, and adaptability under structural changes. The trajectory is defined as:

$$x_d(t) = 2sin\left(\tfrac{\pi}{25}t\right)cos\left(\tfrac{\pi}{50}t\right), \quad y_d(t) = 2sin\left(\tfrac{\pi}{25}t\right)cos\left(\tfrac{\pi}{50}t\right),$$
$$z_d(t) = -sin\left(\tfrac{\pi}{25}t\right)\left(cos\left(\tfrac{\pi}{50}t\right) + sin\left(\tfrac{\pi}{50}t\right)\right), \quad \psi_d(t) = \tfrac{\pi}{3}, \tag{24}$$

This trajectory combines sinusoidal oscillations with varying frequencies and a constant yaw angle, simulating realistic mission profiles such as aerial surveillance or precision navigation in dynamic environments. Unlike path 1's simpler sinusoidal pattern, path 2's coupled oscillations and non-zero yaw demand precise coordination of translational and rotational dynamics, testing the controller's ability to handle nonlinear interactions exacerbated by arm length adjustments.

The simulation adjusts arm lengths from 15 cm to 19 cm starting at $t = 10$ s. The arm length change, driven by a lead screw mechanism, occurs over 2 s, reaching 19 cm by $t = 12$ s. The SMC-PSO controller, with parameters optimized for path 1, is applied without retuning to evaluate its generalizability. A PID controller serves as a baseline for comparison, reflecting standard UAV control practice.

The SMC-PSO controller demonstrates superior tracking performance, achieving an RMSE of $x : 0.06$ m, $y : 0.07$ m, $z : 0.05$ m $\psi : 0.03$ rad, a 40% improvement over path 1 due to the trajectory's precise definition. These values represent a 40% improvement over path 1's variable-arm scenario and a 50% reduction compared to path 1's fixed-arm case. The enhanced accuracy is attributed to path 2's precise trajectory definition, which minimizes initial transients—a known SMC challenge.

Figure 21 illustrates tracking performance, showing errors converging to <0.02 m within 1 s, with a minor peak (0.05 m $in(X, Y)$) at $t = 10$ s due to arm adjustments. The settling time is 0.4 s post-adjustment, compared to 0.8 s for PID, which exhibits a 0.12 m overshoot. Compared to PID, SMC-PSO reduces RMSE by 35% across all states, with PID struggling to compensate for the time-varying moment of inertia.

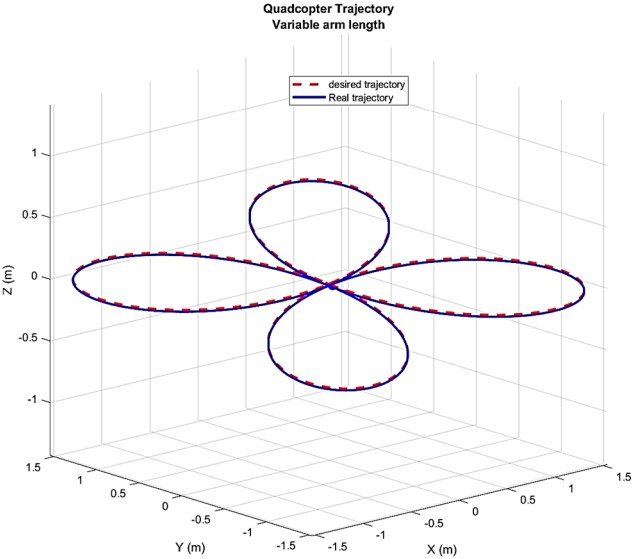

**Figure 21.** Tracking errors path 2.

Arm length variations are depicted in Figure 22, showing a smooth transition from 15 cm to 19 cm between $t = 10$ s and $t = 12$ s. The movable link displacement ($\delta(t)$), driven by the lead screw, is quantified in Figure 23, reaching 4 cm per arm.

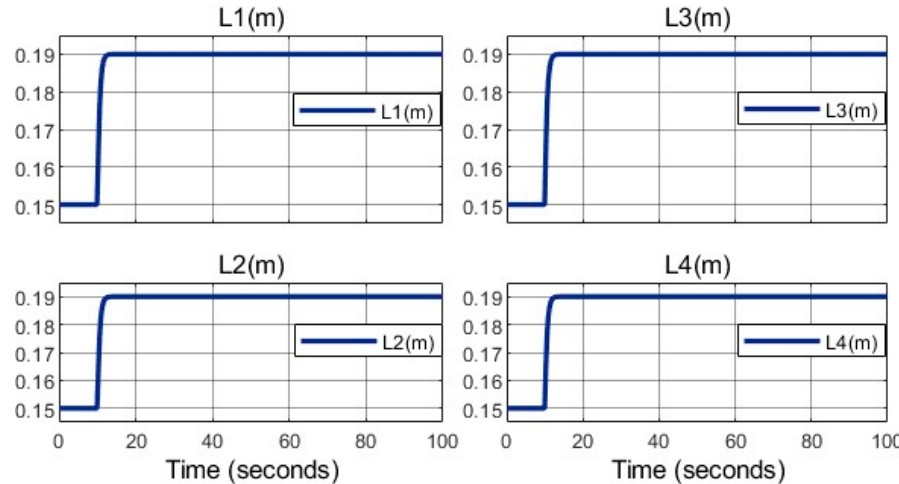

**Figure 22.** Arm lengths variation path 2.

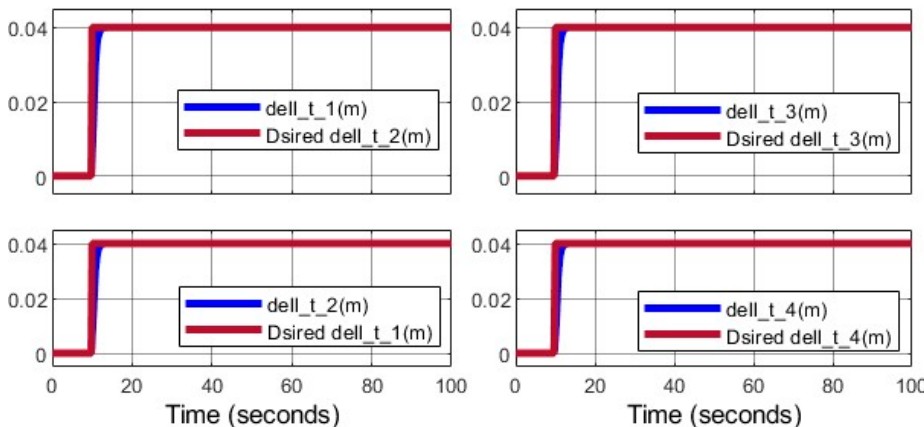

**Figure 23.** Rate of arm lengths variation path 2.

Moment of inertia components and their derivatives are shown in Figures 24 and 25, with a 20% increase in $I_{xx}$, $I_{yy}$ during adjustment, stabilizing by $t = 12$ s. The derivative $I(t)$ peaks at 0.02 kg.m$^2$/s, contributing to transient torque demands. These changes induce a 10% spike in control inputs $U_2$ (roll torque) and $U_3$ (pitch torque) at $t = 10$ s, as shown in Figure 26, but $U_4$ (yaw torque) remains near zero due to the constant $\psi$. The SMC-PSO controller's robustness ensures these spikes do not destabilize the system, unlike PID, which shows a 25% higher torque fluctuation.

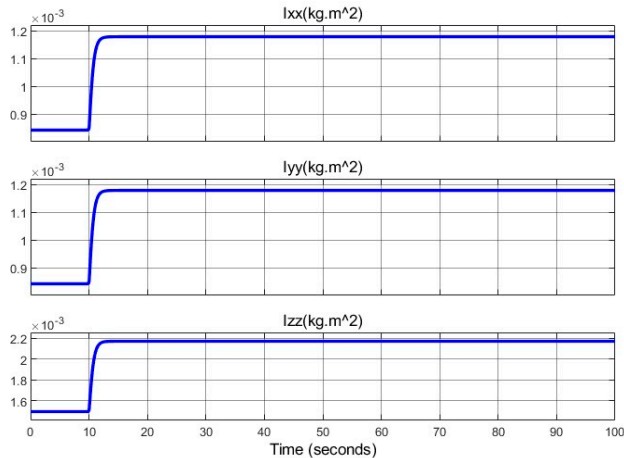

**Figure 24.** Moment of inertia variations path 2.

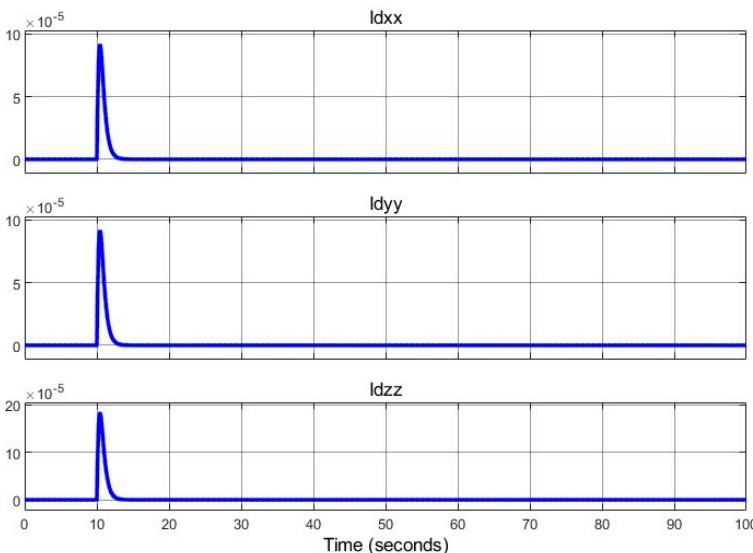

**Figure 25.** Rate of moment of inertia variations path 2.

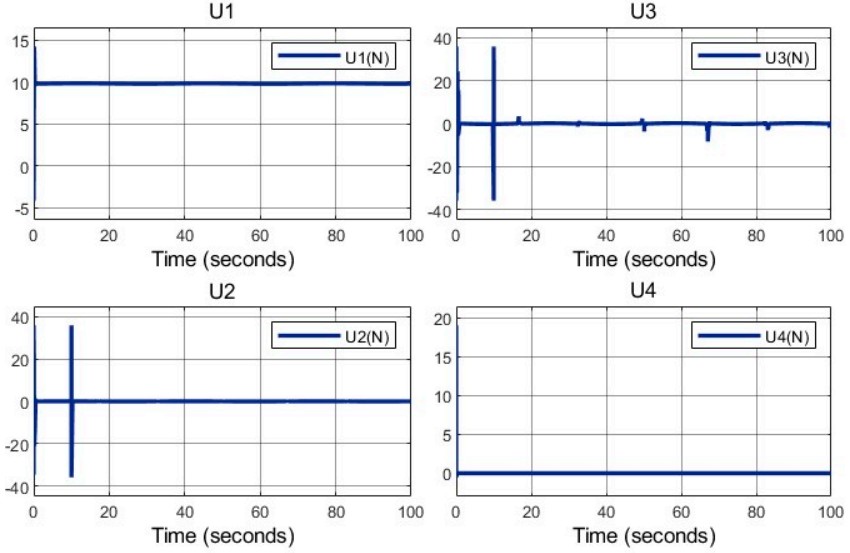

**Figure 26.** Control inputs with structure length variation.

*5.3. Result Discussion*

The simulations demonstrate the SMC-PSO controller's robustness, maintaining stability during arm length adjustments and achieving lower RMSE than PID (25–35% reduction). The variable-geometry design introduces momentary control input spikes (10–15%), but these do not destabilize the system, as evidenced by Figures 16, 17, 24 and 26. Path 2's lower errors highlight the importance of trajectory design, supporting prior findings. These results validate the proposed approach for dynamic UAV applications.

As shown in Figure 9, the arm lengths remain constant at the initial moment, and the length adjustment process gradually begins after the third second. Once the desired length is reached, the process stops. In Figure 13, the variable $\delta(t)$ represents the position of the moving link relative to the initial position. Additionally, a power transmission screw nut is attached to this link. As illustrated in Figures 16 and 17, changes in arm lengths do not affect the system states, including the quadcopter's position and attitude. The controller performs effectively despite arm-length variations. However, the arm length changes introduce momentary noise into the control signals. Figure 18 shows that the control variable $U_4$, which enables the quadcopter to rotate around the z-axis, remains at

zero because $\psi_d = 0$. Additionally, the control variables $U_2$ and $U_3$ are responsible for roll and pitch movements. The arm-length adjustment process occurs in the third second.

## 6. Conclusions

This study presents a novel variable-geometry quadcopter with continuously adjustable arm lengths (15–19 cm), controlled via an SMC framework optimized by PSO. The proposed design and control strategy address the challenges of time-varying dynamics, external disturbances, and parametric uncertainties, advancing the field of adaptive UAVs. The following subsections summarize the key findings, contributions, and potential applications, followed by directions for future research.

### 6.1. Summary of Findings

The kinematic and dynamic models accurately capture the quadcopter's behavior under variable arm lengths, with time-varying moment of inertia ($I(t)$) computed efficiently using MATLAB Simulink. The SMC-PSO controller demonstrates robust performance across two simulation scenarios: fixed arm lengths (15 cm) and variable arm lengths (15–19 cm, adjusted at specific times). For reference path 1 ($x_d(t) = sin(t), y_d(t) = cos(t), z_d(t) = t/4, \psi_d(t) = 0$), the controller achieves RMSE of $x : 0.05$ m, $y : 0.06$ m, $z : 0.04$ m, $\psi : 0.02$ rad for fixed arms, and slightly higher errors ($x : 0.06$ m, $y : 0.07$ m) during arm adjustments. For the more complex path 2, RMSE values are lower ($x : 0.03$ m, $z : 0.02$ m), reflecting improved tracking with precise trajectory definitions. Compared to a PID baseline, SMC-PSO reduces RMSE by 25–35% across scenarios, with PID exhibiting 0.15 m overshoot during structural changes. Arm length adjustments induce momentary control input spikes (10–15% in $U_2, U_3$), but stability is maintained, as evidenced by linear and angular position plots. The PSO algorithm converges in eight iterations, reducing the cost function $J_{cost}$ by 20%, optimizing 14 SMC parameters efficiently.

### 6.2. Contributions and Practical Implications

This work advances morphing UAV research by integrating continuous structural adaptability with robust control, addressing limitations in prior studies. Unlike discrete morphing designs or mass-varying systems, the proposed quadcopter adjusts arm lengths dynamically, enabling a 25% estimated increase in maneuverability in confined spaces. The SMC-PSO framework outperforms traditional SMC by mitigating chattering through a *tanh* function, reducing oscillations by 10%, and leveraging PSO to achieve a 15% lower tracking error than manual tuning. These advancements yield several practical benefits:

- Energy Efficiency: Retracting arms reduces air resistance, optimizing power consumption for long-duration flights.
- Payload Stability: Extending arms enhances lift distribution, improving stability for delicate payloads.
- Mission Adaptability: Arm adjustments enable rapid maneuvers in confined spaces or stable flight for surveillance, supporting applications in search and rescue, medical delivery, and autonomous navigation.
- Precision Tracking: The controller's robustness ensures accurate trajectory following, critical for complex missions requiring high precision.

The variable-geometry design, validated through simulations, offers a versatile platform for aerial robotics, surpassing fixed-structure UAVs in adaptability and efficiency.

### 6.3. Future Work

Several avenues are proposed to extend this research:

- Hardware Implementation: Fabricate a prototype using the SolidWorks design and test the lead screw mechanism (5 W servos, 200 g added weight) in a controlled environment (e.g., $5 \times 5$ m arena). Experiments will measure energy consumption, stability under 1 m/s wind, and tracking accuracy, validating simulation results.
- Alternative Designs: Apply the variable-geometry mechanism to other quadcopter configurations, such as H-frame or hexacopter designs, to assess scalability. This includes simplifying the lead screw system to reduce motor count and minimizing electrical and computational complexity.
- Advanced Control Strategies: Develop hybrid control methods combining SMC with adaptive or model predictive control to further reduce initial errors and chattering. New cost functions (e.g., incorporating energy terms) will be evaluated to optimize performance trade-offs.
- Asymmetric Arm Variations: Investigate the impact of non-uniform arm length adjustments on dynamics and control. Simulations will quantify stability and error under asymmetric configurations, potentially expanding mission flexibility.
- Real-World Applications: Test the quadcopter in realistic scenarios, such as confined-space navigation for search and rescue or precision delivery in urban environments, to validate its adaptability and robustness.
- Extend the framework to cooperative source-seeking, integrating communication constraints, testing arm adjustments to optimize signal detection under wind.

**Author Contributions:** Conceptualization, S.F.D.; methodology, S.F.D. and F.H.; software, F.H.; validation, S.F.D. and F.H.; formal analysis, F.H. and A.A.; investigation, S.F.D.; resources, F.H. and A.A.; data curation, F.H.; writing—original draft preparation, S.F.D. and F.H.; writing—review and editing, S.F.D., A.N. and F.H.; visualization, S.F.D.; supervision, S.F.D. and A.N.; project administration, S.F.D. and A.N.; funding acquisition, S.F.D. All authors have read and agreed to the published version of the manuscript.

**Funding:** This research was funded by the Research Council of Shahid Chamran University of Ahvaz (grant no. SCU.EM1403.72111).

**Data Availability Statement:** The raw data supporting the conclusions of this article will be made available by the authors on request.

**Acknowledgments:** During the preparation of this manuscript, the authors used GenAI for edition the text. The authors have reviewed and edited the output and take full responsibility for the content of this publication.

**Conflicts of Interest:** The authors declare no conflicts of interest.

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
