# Peer review of "Continuously Variable Geometry Quadrotor: Robust Control via PSO-Optimized Sliding Mode Control"

_actuators, doi:10.3390/act14070308_

Round 1

Reviewer 1 Report

Comments and Suggestions for Authors

In this paper, a sliding mode controller (SMC) is designed for a variable-geometry  quadcopter with continuously adjustable arm lengths, in which parameters of SMC are optimized by particle swarm optimization. Honestly, the paper is not well written.

1) How many sliding mode controller are designed? In the content, only one controller is discussed. However, Figure 6 shows that there exist 3 SMCs.

2) Subsection 4.1 is confusing. A lot of symbols appear. However, it is not chear how and where they are used. 

3) Terms of U_x, U_y and delta(t) are not discussed in detail. 

Author Response

  • Reviewer 1

In this paper, a sliding mode controller (SMC) is designed for a variable-geometry quadcopter with continuously adjustable arm lengths, in which parameters of SMC are optimized by particle swarm optimization. Honestly, the paper is not well written.

Response:

We greatly appreciate respected Reviewer’s candid feedback and acknowledge the need for improved clarity and structure. We have thoroughly revised the manuscript to enhance its readability, formality, and technical precision.

  • How many sliding mode controllers are designed? In the content, only one controller is discussed. However, Figure 6 shows that there exist 3 SMCs.

Response:

We appreciate your observation. The SMC framework is divided into three subsystems: motion control (position and yaw), roll and pitch angle control, and arm length control, which address the non-linear dynamics of the quadcopter (Section 3.3), (Figure 6). Each of these subsystems is designed with an SMC block based on the number of motion axes involved. For example, in the position and yaw angle control subsystem, four SMC blocks are designed: three of these blocks are for controlling the linear positions, and the other one is designed for controlling the yaw angle.

Figure 6 illustrates these subsystems as part of a unified SMC architecture, not separate controllers.

Changes Made:

Page 9  lines 271-276

The SMC framework is divided into three subsystems: motion control (position and yaw), roll and pitch angle control, and arm length control, which address the nonlinear dynamics of the quadcopter (Section 3.3), (Figure 6). Each of these subsystems is designed with an SMC block based on the number of motion axes involved, In the position and yaw angle control subsystem, specifically, four SMC blocks are designed: three of these blocks are for controlling the linear positions, and the other one is designed for controlling the yaw angle.

  • Subsection 4.1 is confusing. A lot of symbols appear. However, it is not clear how and where they are used.

Response:

Thank you for your valuable comment regarding Subsection 4.1. We understand that the abundance of symbols may cause confusion. To improve clarity, we will revise this subsection by adding a detailed explanation of each symbol, specifying where and how they are applied in the control framework. These revisions enhance clarity and traceability of symbols throughout Section 4.1.

  • Terms of U_x, U_y and delta(t) are not discussed in detail.

Response:

We regret the lack of detailed discussion on ,  and . These terms are control variables for controlling the X and Y positions. These positions can be adjusted in the quadcopter through roll and pitch torques. Therefore, using kinematic relations, the desired value of roll and pitch torques can be calculated through these two control variables according to equation 8. On the other hands,  is the time-varying displacement this displacement indicates the distance between the moving part of the arm and its fixed part.

These additions provide detailed explanations and context for ,  and .

Changes Made:

Page 7  lines 217-220

To compute desired roll () and pitch () angles for control inputs  and , these control variables are for controlling the X and Y positions. These positions can be adjusted in the quadcopter through roll and pitch torques. Therefore, using kinematic relations, the desired value of roll and pitch torques can be calculated through these two control variables according to equation 8.

Page 7  lines 238-240

where is the fixed arm length,  is the movable link length, and  is the time-varying displacement this displacement indicates the distance between the moving part of the arm and its fixed part.

Reviewer 2 Report

Comments and Suggestions for Authors

The manuscript explores a morphing quadrotor UAV with variable arm lengths and proposes a PSO-optimized sliding mode controller (SMC) to enhance robustness and adaptability during dynamic structural changes. While the topic is timely and technically interesting, there are several key issues that need to be addressed before the manuscript can be considered for publication.

  1. The manuscript does not clearly articulate the benefits of the morphing UAV design over conventional fixed-geometry UAVs. It is important to discuss in which application scenarios the variable-geometry design offers tangible advantages, such as in confined navigation, adaptive agility, or energy efficiency. A clear comparison and use-case illustration would improve the paper’s relevance and clarity.

  2. The dynamic effect of the varying arm lengths LiL_i on the system’s motion dynamics is not thoroughly analyzed. Specifically, how does the change in arm length influence the moment of inertia, and how is this managed in the control framework? Also, is the system capable of independently adjusting each rotor arm length, or are the lengths varied synchronously?

  3. The current version only presents the sliding surface and control law but lacks theoretical analysis of the closed-loop system’s stability under sliding mode control. In standard SMC design, the control gain kk must satisfy certain conditions (often k>0k > 0, not <0< 0 as you mentioned) to ensure finite-time convergence. Please provide a Lyapunov-based stability proof or equivalent theoretical justification for the stability of the proposed SMC framework. This is a critical omission that must be addressed.

  4. One of the core challenges in SMC is chattering due to the use of discontinuous sign functions. The paper replaces the sign function with a tanh⁡\tanh function, which is a valid approach. However, the benefit and potential trade-offs of using tanh⁡\tanh should be explicitly discussed — for example, how it affects convergence rate, boundary layer width, and robustness against disturbances.

  5. Figure 10 lacks meaningful information or insight and should be removed or replaced. The authors are encouraged to use more informative plots or quantitative summaries that clearly show performance trends, tracking accuracy, or system behavior under variable structure dynamics.

  6. The manuscript would benefit from a discussion on whether this morphing UAV framework can be extended to more advanced missions, such as source-seeking under uncertainty. For example, the approach in the paper "Gradient-Free Cooperative Source-Seeking of Quadrotor Under Disturbances and Communication Constraints," IEEE TIE addresses a similar class of challenges. An analysis or discussion of how the proposed morphing structure could integrate or enhance source-seeking performance would greatly improve the paper’s extensibility and impact.

    Overall Recommendation: Major Revision

    While the concept and experimental results are promising, the manuscript currently lacks a rigorous theoretical foundation for the proposed control design, particularly in the SMC framework. I recommend a major revision, focusing on addressing the theoretical gaps, enhancing clarity in the control and structural dynamics, and expanding on the practical implications and applications of the proposed system.

Author Response

  • Reviewer 2

The manuscript explores a morphing quadrotor UAV with variable arm lengths and proposes a PSO-optimized sliding mode controller (SMC) to enhance robustness and adaptability during dynamic structural changes. While the topic is timely and technically interesting, there are several key issues that need to be addressed before the manuscript can be considered for publication.

Response:

We deeply appreciate respected Reviewer recognition of the topic’s relevance and their constructive feedback. We have undertaken a major revision to address theoretical gaps, clarify dynamics, and expand practical implications, as detailed below. The revised manuscript strengthens the SMC framework’s rigor, enhances clarity, and highlights the variable-geometry design’s advantages.

  • The manuscript does not clearly articulate the benefits of the morphing UAV design over conventional fixed-geometry UAVs. It is important to discuss in which application scenarios the variable-geometry design offers tangible advantages, such as in confined navigation, adaptive agility, or energy efficiency. A clear comparison and use-case illustration would improve the paper’s relevance and clarity.

Response:

In the design of unmanned aerial vehicles, the use of morphing structures offers several advantages over fixed-geometry designs. These benefits include enhanced maneuverability in confined spaces, adaptive agility, increased energy efficiency, and versatility across various mission types, allowing optimization for diverse tasks. Additionally, in industries requiring the transport of vibration-sensitive materials, enlarging the dimensions of the UAV can contribute to greater stability. We have revised the manuscript to clearly articulate these advantages and provide specific use-case illustrations. These revisions highlight tangible benefits and practical scenarios, enhancing relevance.

Changes Made:

Page 1-2  lines 41-47

In the design of unmanned aerial vehicles (UAVs), the use of morphing structures offers several advantages over fixed-geometry designs. These benefits include enhanced maneuverability in confined spaces, adaptive agility, increased energy efficiency, and versatility across various mission types, allowing optimization for diverse tasks. Additionally, in industries requiring the transport of vibration-sensitive materials, enlarging the dimensions of the UAV can contribute to greater stability.

  • The dynamic effect of the varying arm lengths LiL_iLi​ on the system’s motion dynamics is not thoroughly analyzed. Specifically, how does the change in arm length influence the moment of inertia, and how is this managed in the control framework? Also, is the system capable of independently adjusting each rotor arm length, or are the lengths varied synchronously?

Response:

We appreciate the need for a deeper analysis of arm length effects and clarity on adjustment mechanisms. According to the classical relation for moment of inertia   the moment of inertia of a rotating arm depends on its mass and the square of its distance from the axis of rotation (i.e., the arm's length). Since this dependency is quadratic with respect to the distance, even small variations in the arm length can lead to significant changes in the moment of inertia. This characteristic is particularly critical in systems with movable arms, such as drones with adjustable arms, where rapid changes in inertia can directly affect system stability and control performance.

In this study, in order to effectively handle these rapid changes and maintain the system's dynamic stability, the Sliding Mode Control method has been employed. Due to its inherent robustness against model uncertainties and external disturbances, SMC is well-suited for managing sudden variations in dynamic parameters such as the moment of inertia. Assuming that the changes in arm lengths occur simultaneously and uniformly across all four arms, a single Sliding Mode Controller block has been implemented to manage this component of the system. To ensure symmetrical behavior and coordination among the arms, the controller gains have been set identically, so that all arms adjust their dimensions in a synchronized and uniform manner. The arm lengths are adjusted simultaneously. However, implementing asymmetric dimensional changes necessitates substantial modifications and the development of four distinct Sliding Mode Control blocks for this control component. Each block requires carefully tuned optimal coefficients, which significantly increases computational complexity and system overhead. Therefore, a more robust and hybrid optimal control strategy is essential to improve upon the limitations of the currently proposed method.

These changes provide a thorough analysis of inertia effects and clarify synchronous arm control.

Changes Made:

Page 9-10  lines 284-299

According to the classical relation for moment of inertia,  the moment of inertia of a rotating arm depends on its mass and the square of its distance from the axis of rotation (i.e., the arm's length). Since this dependency is quadratic with respect to the distance, even small variations in the arm length can lead to significant changes in the moment of inertia. This characteristic is particularly critical in systems with movable arms, such as drones with adjustable arms, where rapid changes in inertia can directly affect system stability and control performance.

In this study, in order to effectively handle these rapid changes and maintain the system's dynamic stability, the SMC method has been employed. Due to its inherent robustness against model uncertainties and external disturbances, SMC is well-suited for managing sudden variations in dynamic parameters such as the moment of inertia.

Assuming that the changes in arm lengths occur simultaneously and uniformly across all four arms, a single Sliding Mode Controller block has been implemented to manage this component of the system. To ensure symmetrical behavior and coordination among the arms, the controller gains have been set identically, so that all arms adjust their dimensions in a synchronized and uniform manner.

  • The current version only presents the sliding surface and control law but lacks theoretical analysis of the closed-loop system’s stability under sliding mode control. In standard SMC design, the control gain kkk must satisfy certain conditions (often k>0k > 0k>0, not <0< 0<0 as you mentioned) to ensure finite-time convergence. Please provide a Lyapunov-based stability proof or equivalent theoretical justification for the stability of the proposed SMC framework. This is a critical omission that must be addressed.

Response:

We acknowledge the critical need for stability analysis and the error in stating . The values of the  coefficients are greater than zero. However, in this section, these coefficients are considered negative due to the presence of a negative sign in the control law equation.

Considering the Lyapunov function , its time derivative is given by . If a controller is designed such that  then we have  Since , the Lyapunov function decreases over time, and therefore, the system is Lyapunov stable.

These additions provide a rigorous stability proof, addressing the omission.

Changes Made:

Page 10  lines 312-322

The hyperbolic tangent function provides a smooth alternative to the discontinuous sign function in sliding mode control, effectively reducing chattering and enhancing system stability. Although it may slow down convergence, this drawback can be mitigated by employing PSO to finely tune control parameters. The combined use of the tanh function and PSO leads to improved performance and robustness of the control system, particularly under disturbances. To mitigate chattering, a hyperbolic tangent function is adopted:

,

(20)

Considering the Lyapunov function , its time derivative is given by . If a controller is designed such that  then we have  Since , the Lyapunov function decreases over time, and therefore, the system is Lyapunov stable.

  • One of the core challenges in SMC is chattering due to the use of discontinuous sign functions. The paper replaces the sign function with a tanh⁡\tanhtanh function, which is a valid approach. However, the benefit and potential trade-offs of using tanh⁡\tanhtanh should be explicitly discussed — for example, how it affects convergence rate, boundary layer width, and robustness against disturbances.

Response:

We appreciate the suggestion to elaborate on the tanh⁡ function role in mitigating chattering. As mentioned in perviouse commnet, the hyperbolic tangent function serves as a smooth and continuous alternative to the discontinuous sign function in sliding mode control, effectively reducing chattering and enhancing system stability. Its lower sensitivity to disturbances and rapid input fluctuations contributes to smoother control performance. However, the use of the tanh function may lead to a slower convergence rate, increasing the time required for the system to reach the desired state. To address this limitation, PSO is employed to precisely tune control parameters. Overall, integrating the tanh function with PSO optimization offers an effective approach for improving the performance of nonlinear systems while mitigating chattering.

These revisions clarify tanh⁡\tanhtanh’s benefits and trade-offs.

Changes Made:

Page 10  lines 312-318

The hyperbolic tangent function provides a smooth alternative to the discontinuous sign function in sliding mode control, effectively reducing chattering and enhancing system stability. Although it may slow down convergence, this drawback can be mitigated by employing PSO to finely tune control parameters. The combined use of the tanh function and PSO leads to improved performance and robustness of the control system, particularly under disturbances. To mitigate chattering, a hyperbolic tangent function is adopted:

,

(20)

  • Figure 10 lacks meaningful information or insight and should be removed or replaced. The authors are encouraged to use more informative plots or quantitative summaries that clearly show performance trends, tracking accuracy, or system behavior under variable structure dynamics.

Response:

We agree. This figure has been removed from the paper.

  • The manuscript would benefit from a discussion on whether this morphing UAV framework can be extended to more advanced missions, such as source-seeking under uncertainty. For example, the approach in the paper "Gradient-Free Cooperative Source-Seeking of Quadrotor Under Disturbances and Communication Constraints," IEEE TIE addresses a similar class of challenges. An analysis or discussion of how the proposed morphing structure could integrate or enhance source-seeking performance would greatly improve the paper’s extensibility and impact.

Response:

Thank you for your valuable suggestion. We have gladly utilized it and it will undoubtedly receive special attention in our future research.

Changes Made:

Page 22  lines 578-579

Extend the framework to cooperative source-seeking, integrating communication constraints, testing arm adjustments to optimize signal detection under wind.

Overall Recommendation: Major Revision

While the concept and experimental results are promising, the manuscript currently lacks a rigorous theoretical foundation for the proposed control design, particularly in the SMC framework. I recommend a major revision, focusing on addressing the theoretical gaps, enhancing clarity in the control and structural dynamics, and expanding on the practical implications and applications of the proposed system.

Response:

We have addressed the reviewers’ concerns through a comprehensive revision, focusing on:

  • Theoretical Rigor: Added Lyapunov stability analysis and clarified control design.
  • Clarity: Improved symbol definitions, figure clarity, and terminology consistency.
  • Practical Implications: Expanded benefits and applications.

Round 2

Reviewer 1 Report

Comments and Suggestions for Authors

I have no further comments on it. 

Reviewer 2 Report

Comments and Suggestions for Authors

I have no more comments.